# ARL6IP5 Ameliorates α-Synuclein Burden by Inducing Autophagy via Preventing Ubiquitination and Degradation of ATG12

**DOI:** 10.3390/ijms241310499

**Published:** 2023-06-22

**Authors:** Ibrar Siddique, Kajal Kamble, Sakshi Gupta, Kavita Solanki, Sumnil Bhola, Nuzhat Ahsan, Sarika Gupta

**Affiliations:** Molecular Science Laboratory, National Institute of Immunology, New Delhi 110067, India

**Keywords:** ARL6IP5 (ADP-ribosylation-like factor 6 interacting protein 5), Parkinson’s disease (PD), autophagy, neurodegeneration, α-synuclein, SH-SY5Y cells

## Abstract

Recent advanced studies in neurodegenerative diseases have revealed several links connecting autophagy and neurodegeneration. Autophagy is the major cellular degradation process for the removal of toxic protein aggregates responsible for neurodegenerative diseases. More than 30 autophagy-related proteins have been identified as directly participating in the autophagy process. Proteins regulating the process of autophagy are much more numerous and unknown. To address this, in our present study, we identified a novel regulator (ARL6IP5) of neuronal autophagy and showed that the level of ARL6IP5 decreases in the brain with age and in Parkinson’s disease in mice and humans. Moreover, a cellular model of PD (Wild type and A53T mutant α-synuclein overexpression) has also shown decreased levels of ARL6IP5. ARL6IP5 overexpression reduces α-synuclein aggregate burden and improves cell survival in an A53T model of Parkinson’s disease. Interestingly, detailed mechanistic studies revealed that ARL6IP5 is an autophagy inducer. ARL6IP5 enhances Rab1-dependent autophagosome initiation and elongation by stabilizing free ATG12. We report for the first time that α-synuclein downregulates ARL6IP5 to inhibit autophagy-dependent clearance of toxic aggregates that exacerbate neurodegeneration.

## 1. Introduction

Parkinson’s disease (PD) is the second most fatal neurodegenerative disorder, identified by neuronal degeneration in the substantia nigra pars compacta (SNpc) and intracellular deposition of Lewy bodies [1]. The most common symptoms of PD are divided into motor and non-motor symptoms. Motor symptoms include mainly tremor, bradykinesia, and rigidity, while non-motor symptoms consist of rapid eye movement, sleeping disorder, olfactory dysfunction, anosmia, constipation, urinary dysfunction, cognitive impairment, and depression [2]. There is still no clear understanding of the etiology of Parkinson’s disease, despite decades of vigorous research. As existing treatments provide only symptomatic relief, there is a high need for therapeutic strategies to reduce the accumulated α-synuclein (SNCA) level in neurons. Neurons are post-mitotic cells that highly depend on the basal autophagy to maintain their proteostasis.

The ubiquitin proteasomal system (UPS) and autophagy are the two main protein degradation pathways of eukaryotes. UPS degrades short-lived soluble proteins, while autophagy degrades long-lived insoluble aggregates [3]. As a result of autophagy, cells survive adverse conditions, such as nutrient deprivation, the accumulation of aggregated proteins, the emergence of damaged organelles, and the recycling of receptors. Autophagy plays a pivotal role in maintaining protein homeostasis. Dysfunctional autophagy was identified as the main cause of many neurodegenerative disorders [4]. The process of autophagy can be divided into mechanistically distinct steps, which include induction, cargo recognition and selection, autophagosome formation, autophagosome-lysosome fusion, and breakdown of the cargo followed by release of the degradation products in the form of small peptides back into the cytoplasm [5]. The 1990s saw an era of autophagy research, leading to the characterization of this phenomenon at the molecular level [6]. In 1993, Tsukada and Ohsumi performed the isolation and characterization of 15 *S. cerevisiae* mutants that displayed defective autophagy and named them *apg1*-*15* (autophagy) [7]. Further research was done to find the more defective mutants by other researchers. In 2003, a unified nomenclature for the so-called autophagy-related genes/proteins, ATGs, was proposed [8]. Recently, yeast ATG39 and ATG40 have been identified as receptors that are apparently involved in the selective removal of the cytoplasmic and perinuclear ER and the nucleus [9]. Most of the yeast ATG proteins have mammalian homologs [10]. Different sets of ATG proteins are involved in these steps and comprise the core autophagic machinery. Besides the ATG proteins, certain non-ATG proteins are required for autophagy, such as certain specific Sec-proteins that are important for autophagosome formation [11]. Moreover, Trs85 is another protein that is a component in the TRAPP (transport protein particle) complexes, which mediate ER-to-Golgi and intra-Golgi trafficking and are found in both nonselective autophagy and selective autophagy [12]. The number of such autophagy-regulating proteins is increasing with research.

ADP-ribosylation-like factor 6 interacting protein 5 (ARL6IP5) is also known as JWA in humans, Addicsin in mice, and GTRAP 3-18 or JM4 in rats [13]. This protein belongs to the PRAF3 family as it has a functionally large prenylated acceptor domain 1, mainly involved in intracellular protein trafficking [14]. ARL6IP5 is an established negative regulator of the EAAC1 transporter [15]. This protein has been extensively studied in the field of cancer metastasis, yet the functions of ARL6IP5 are less well characterized [16,17,18]. In a small study, siRNA ARL6IP5 expression reduced autophagy in cancer cells [19]. In the present study, we identified ARL6IP5 as an ATG12 interacting protein and a novel regulator/inducer of autophagy that could reduce α-synuclein aggregates in situ. Our detailed findings thereby establish the critical role of the ARL6IP5/Rab1/ATG12 axis for neuroprotection in PD.

## 2. Results

### 2.1. The Level of ARL6IP5 Decreases with Age and Parkinson’s Disease

To study the role of ARL6IP5 in aging and neurodegenerative diseases, we checked the age-dependent (4, 8, and 12 months) change in the level of ARL6IP5 in the wild-type C57BL6 mouse brain (WT) and PD transgenic mouse model (A53T mutant) (Tg) (Figure 1A,B). We observed a decrease in ARL6IP5 level with aging in both WT and Tg mice. For WT mice, there was a decrease 25 ± 7%, *p* = 0.12, *n* = 3 and 44 ± 17%, *p* = 0.0015, *n* = 3 at the age of 8 months and 12 months, respectively, in the ARL6IP5 level as compared to 4 months. Tg mice had a trend showing lower levels of ARL6IP5 as compared to wild-type controls for all time points but with high *p*-value (Figure 1A,B). Data clearly suggest that ARL6IP5 levels decline with aging, which shows a synergic effect with α-synuclein overexpression.

To revalidate our immunoblot data, we performed immunofluorescence in the brain sections of the WT and Tg mice models of PD at 12 months of age. As shown in Appendix A, the α-synuclein aggregate burden was higher (56 ± 19%, *p* = 0.0039, *n* = 3) in the Tg mouse model of PD compared to WT and agrees with the disease burden, and ARL6IP5 was decreased (71 ± 14%, *p* = 0.0002, *n* = 3) as compared to WT. After confirming the disease burden histologically in mice, we detected the level of ARL6IP5 in the human brain sections and were excited to see the decrease (85 ± 28%, *p* = 0.0042, *n* = 3) in its level in the PD patients’ brain samples as compared to healthy controls (Figure 1C,D). Additionally, we assessed the level of ARL6IP5 in the brain lysates of 6-OHDA-induced Parkinson’s disease in mice. 6-hydroxydopamine (6-OHDA) is a synthetic organic neurotoxin. Injection of the neurotoxin, 6-OHDA in substantia niagra of brain reproduces several non-motor comorbidities commonly associated with PD, including cognitive deficits, depression, anxiety and deposition of α-synuclein in the brain [20,21]. C57BL6 mice were intracranially injected with various doses of 6-OHDA stereotaxically or injected with vehicle to further validate our findings. Interestingly, the level of ARL6IP5 significantly decreases (20 ± 11% *p* = 0.75, *n* = 3, 60 ± 23%, *p* = 0.0009, *n* = 3, and 75 ± 15% *p* < 0.0001, *n* = 3 at the doses of 30 µM, 40 µM, and 50 µM, respectively) in the 6-OHDA-induced mouse model of PD as compared to the sham control (Figure 1E,F).

To further understand the role of ARL6IP5 in neurodegeneration, we developed a cellular model of PD by stably expressing wild-type and A53T mutant α-synuclein in the SHSY5Y neuronal cell line. In accordance with animal data, there was a significant reduction (80 ± 24%, *p* < 0.0001, *n* = 3) in ARL6IP5 levels in cells overexpressing α-synuclein and reduction (60 ± 25%, *p* < 0.0001, *n* = 3) in A53T mutant α-synuclein overexpression compared to control cells (Figure 1I,J). SH-SY5Y cells stably expressing mutant GFP-A53T α-synuclein were transfected with ARL6IP5, and the level of green fluorescence (as a measure of expression of A53T α-synuclein) was measured. We found a decrease (28 ± 33 in ARL6IP5 transfected cells from 58 ± 24 in control cells, *p* < 0.0001) in the level of A53T-α-synuclein fluorescence with ARL6IP5 expression (Figure 1K,L). This indicates a direct role of ARL6IP5 in aging and PD-like neurodegenerative diseases.

### 2.2. Beneficial Role of ARL6IP5 Overexpression in PD

Having learned that α-synuclein (WT/A53T) overexpression reduces ARL6IP5 levels, we wanted to check the change in toxicity and aggregate of α-synuclein (WT/A53T) when the level of ARL6IP5 is overexpressed. The toxicity of cellular aggregate was checked using the A11 antibody as previously described [22]. We were delighted to detect a significant reduction in A11 reactivity (31 ± 15%, *p* = 0.49, *n* = 3 in α-synuclein, and 20 ± 16%, *p* = 0.47, *n* = 3 in A53T) with ARL6IP5 overexpression as compared to the basal ARL6IP5 condition (Figure 2A,B). To further confirm the beneficial role of ARL6IP5, we carried out siRNA-mediated knockdown of ARL6IP5 in a cellular model of PD. The toxicity was confirmed with the LDH assay, α-synuclein overexpression in the knockdown condition of ARL6IP5 shows more (15 ± 7%, *p* = 0.018, *n* = 6) toxicity than α-synuclein overexpression alone (Figure 2C).

Till now, our study has confirmed the beneficial role of ARL6IP5 in preventing disease pathogenesis by reducing the burden and toxicity of α-synuclein aggregates. Thus, we proceeded to investigate the molecular mechanisms that ensure cell viability. Subsequently, we planned to investigate the underlying pathways. We studied the effect of ARL6IP5 on cell differentiation and surface receptor tyrosine kinase. The microscopic studies have shown that overexpression of ARL6IP5 induces neuronal differentiation, which was further confirmed by neurite length, which was found to increase (3 ± 2 μm in normal condition to 13 ± 5 µm in ARL6IP5 overexpression condition, *p* < 0.0001, *n* = 20), while α-synuclein overexpression reduced neurite length, which was again restored after co-expression of ARL6IP5 (2 ± 1 μm in α-synuclein overexpression and 11 ± 4 µm in overexpression condition, *p* < 0.0001, *n* = 20) (Figure 2D,E). The role of TAM receptors in neuronal survival and pathological protein aggregate clearance has been suggested [23]. We decided to check using Receptor tyrosine kinase (RTK) array analysis, which revealed a significant increase in the phosphorylation of the TAM receptor family kinases Axl and Mer (Figure 2F–H) in ARL6IP5 overexpression. For Axl, α-synuclein overexpression reduced (54 ± 11%, *p* < 0.0001, *n* = 3 as compared to control) its phosphorylation, which was increased (171 ± 32%, *p* < 0.0001, *n* = 3 as compared to control) in the ARL6IP5 overexpressing condition, and phosphorylation was restored (72 ± 35%, *p* = 0.056, *n* = 3 as compared to control) in co-expression. For Mer, α-synuclein overexpression reduced (74 ± 15%, *p* = 0.097, *n* = 3 as compared to control) its phosphorylation, which was increased (321 ± 24%, *p* < 0.0001, *n* = 3 as compared to control) in ARL6IP5 overexpressing condition, and phosphorylation was restored (110 ± 34%, *p* = 0.97, *n* = 3 as compared to control) in co-expression. Together, all these results suggest that ARL6IP5 restores the cellular conditions that were altered by the overexpression of α-synuclein.

Next, we investigated the plausible cellular degradation pathways involved in the removal of α-synuclein cytosolic aggregates. We observed that overexpression of ARL6IP5 does not significantly affect the proteasomal degradation pathway (Appendix A). In the case of α-synuclein, chaperon-mediated degradation plays a pivotal role in reducing neurotoxicity [24], hence, we studied the levels of p62 and HSC-70 (Appendix A). However, no significant change in the levels of these chaperones was observed upon ARL6IP5 overexpression. Therefore, we ventured into studying the effect of ARL6IP5 overexpression on the cellular bulk degradation process, autophagy. LC3BI is a soluble protein and remains in the cytoplasm. During the formation of autophagosome, LC3BI converts into LC3BII and is incorporated into the membrane of the autophagosome. We measured autophagy by analyzing the level of LCBII, which is a standard marker of autophagy [25]. As shown in Figure 2I,J, the overexpression of α-synuclein disrupted/inhibited (20 ± 8%, *p* = 0.48, *n* = 3, as compared to control) the process of autophagy significantly, which was also inhibited (45 ± 13%, *p* = 0.002, *n* = 3 as compared to control) by the knockdown of ARL6IP5, and shows a synergic effect in inhibition of autophagy (51 ± 23%, *p* = 0.002, *n* = 3), when α-synuclein is overexpressed, and ARL6IP5 is knockdown. Results strongly suggest that ARL6IP5 increases autophagy, so we decided to overexpress ARL6IP5 in α-synuclein overexpression condition and see if it ameliorates the α-synuclein pathology condition via autophagy. Overexpression of ARL6IP5 increased (150 ± 54%, *p* = 0.108, *n* = 3 as compared to control) autophagy. Autophagy was also increased (135 ± 23%, *p* = 0.243, *n* = 3 as compared to control) by ARL6IP5 in α-synuclein overexpressing condition, which was decreased (80 ± 23%, *p* = 0.99, *n* = 3 as compared to control) in α-synuclein overexpressing condition only. We also checked for the autophagy-related protein ATG12, which was decreased (60 ± 24%, *p* = 0.54, *n* = 3) in α-synuclein overexpressing condition and elevated (126 ± 30%, *p* = 0.96, *n* = 3 as compared to control) in ARL6IP5 overexpressing condition, which was normalized (92 ± 34%, *p* = 0.99, *n* = 3 as compared to control) when α-synuclein was co-expressed with ARL6IP5. Rab8 also has a role in autophagosome membrane maturation [26], so we also checked the level of Rab8. Rab8 followed the same pattern as ATG12 and was reduced (55 ± 25%, *p* = 0.27, *n* = 3) in α-synuclein overexpressing condition, upregulated (177 ± 45%, *p*= 0.0005, *n* = 3 as compared to control) in the ARL6IP5 overexpressing condition and remained upregulated (155 ± 46%, *p* = 0.048, *n* = 3 as compared to control) when ARL6IP5 and α-synuclein co-expressed together (Figure 2K,L). These data strongly support that ARL6IP5 upregulates autophagy, which needs to be investigated further.

### 2.3. ARL6IP5 Induces Autophagy Comparable to Other Standard Chemical Inducers in SH-SY5Y Cells

We performed a detailed study to understand the mechanism of ARL6IP5-induced autophagy in the neuron. Firstly, we compared the ARL6IP5-induced autophagy with other autophagy inducers. SH-SY5Y cells were grown to 60% confluency and transfected with 5 μg of the flag-ARL6IP5 construct for 36 h. For another method of autophagy induction, we used serum starvation for 2 h [27], rapamycin (1 μΜ) for 2 h [28], and MeβCD (100 μm) for 24 h [29,30]. ARL6IP5 overexpression increased 261 ± 11%, *p* < 0.0001 the level of LC3BII by, while serum starvation induced 162 ± 62%, *p* = 0.107, Rapamycin induced 254 ± 45%, *p* = 0.74 and MeβCD induced 167 ± 7%, *p* = 0.365 as compared to control. (Figure 3A,B). We also confirmed autophagy induction by ARL6IP5 using GFP-LC3BI [31], which is a soluble cytoplasmic protein. When autophagy is induced, GFP-LC3BI converts into GFP-LC3BII and becomes part of the autophagosome membrane, resulting in the formation of a fluorescence puncta [32]. We counted the number of autophagic puncta in randomly selected ~25 cells from each group. ARL6IP5 overexpressing cells found to have 32 ± 6, *p* < 0.0001 puncta, serum starvation has an average of 30 ± 5, *p* < 0.0001 puncta, rapamycin produced the 30 ± 6, *p* < 0.0001 puncta, MeβCD induced the number of puncta to 29 ± 5, *p* < 0.0001 as compared to control cells, showed an average of 8 ± 2.2 (Figure 3C,D). These findings suggest that ARL6IP5 overexpression induces autophagy, which is comparable in level to the autophagy induced with other standard methods. This suggests that ARL6IP5 may have a role in autophagy regulation, prompting us to investigate further detailed mechanisms.

For our next experiment, we used an empty vector as the transfection control to rule out the possibility of autophagy induction by the transfection reagent. We found a 195 ± 25%, *p* = 0.0001 increase in autophagy measured by the level of LC3BII after ARL6IP5 transfection. Only a marginal induction of 111 ± 10%, *p* = 0.46 percent, was observed after the transfection of empty vector as compared to control (Figure 3E,F). We further wanted to confirm the role of ARL6IP5 in autophagy by downregulating its expression. We transfected cells with siRNA of ARL6IP5 and confirmed the downregulation of protein to 16 ± 3 %, *p* < 0.0001 after 48 h using Western blot technique. When we re-probed the blot to see the level of autophagy using LC3BII, it was reduced to 58 ± 5%, *p* < 0.0001, as compared to control. (Figure 3G,H). We confirmed the phenomenon using GFP-LC3BI and found that the number of autophagy puncta was 8 ± 2 per cell in normal condition. The number of puncta per cell was increased to 32 ± 6, *p* < 0.0001 after ARL6IP5 transfection and reduced less than normal to 3 ± 1, *p* = 0.47 puncta per cell under ARL6IP5 knockdown condition as compared to control (Figure 3I,J).

Another critical question here is whether the autophagy induced by ARL6IP5 overexpression is typical and whether it causes bulk autophagic flux. To address this question, we used the mCherry-GFP-LC3BI vector, which allows us to track the different stages of autophagy [33]. In the absence of autophagy, RFP-GFP fluorescence in the cytoplasm results in combined yellow fluorescence. When autophagy is induced, this protein is incorporated into the autophagosome membrane, forming a yellow punctum. When the autophagosome fuses with the lysosome, which has a low pH, the GFP protein is denaturized, and the green fluorescence is lost. The formation of red puncta indicates the fusion of the autophagosome with the lysosome, and the loss of fluorescence indicates the degradation of autophagosome cargo by autophagy (Figure 3K).

### 2.4. ARL6IP5 Is a Downstream Autophagy Regulator

Next, we wanted to know whether ARL6IP5 overexpression induces autophagy to get rid of the toxic aggregates formed in the cell or if it is part of the typical autophagy induction process. We used standard chemical inducers to induce autophagy and measured the level of ARL6IP5 as autophagy increased. After treatment with rapamycin, serum starvation, MβCD and trehalose, the levels of LC3BII increased by 212 ±7, *p* < 0.0001, 265 ± 10, *p* < 0.0001, 169 ± 14, *p* < 0.0001 and 165 ± 12, *p* < 0.0001 percent, respectively, as compared to control. While the level of ARL6IP5 was also increased to 159 ± 8, *p* < 0.0001, 138 ± 6, *p* < 0.0001, 164 ± 2, *p* < 0.0001 and 133 ± 6, *p* < 0.0001 percent as compared to control, respectively (Figure 4A,B). We further wanted to check the level of autophagy induction using a chemical inducer in the ARL6IP5 knockdown condition. After ARL6IP5 suppression using siRNA, we induced autophagy with serum starvation, rapamycin, MeβCD, and trehalose and found 50 ± 3%, *p* < 0.0001, 76 ± 2%, *p* < 0.0001, 63 ± 0.2%, *p* < 0.0001 and 68 ± 3%, *p* < 0.0001 percent reductions in LC3BII levels as compared to control autophagy induction (Figure 4C,D). A significant decrease in autophagy level with autophagy inducer in ARL6IP5 knockdown conditions strongly suggests that ARL6IP5 is involved in the regulation of the process of autophagy.

### 2.5. ARL6IP5 Overexpression Increase ATG12 by Preventing Its Ubiquitination

The results strongly suggest that ARL6IP5 plays a role in the regulation of autophagy, making it worthwhile to investigate the mechanism by which ARL6IP5 induces autophagy. The first step was to check for the levels of other autophagy-related proteins in the ARL6IP5 overexpressed and downregulated conditions. Surprisingly, the only autophagy-related protein upregulated by ARL6IP5 overexpression was ATG12, which increased by 227 ± 57%, *p* < 0.0001 as compared to control (Figure 4E,F), while ATG5, which works with ATG12 in autophagic membrane propagation [34], remained unchanged (Figure 4E,F). The combined ATG5-ATG12 band was also increased by 308 ± 14%, *p* < 0.0001 with ARL6IP5 overexpression as compared to control, which was contributed solely by ATG12 (Figure 4E,F). ARL6IP5 and ATG12 expression were also checked by knocking down ARL6IP5, and as anticipated, the level of free ATG12 was drastically reduced to 15 ± 3%, *p* < 0.0001 (Figure 4G,H). There was no significant difference observed in the ATG5 and ATG5 + ATG12 levels.

To further understand the relationship between ARL6IP5 and ATG12, we decided to co-immunoprecipitate ARL6IP5 after overexpressing it for 24 h. We found that ATG5 and ATG12 co-immunoprecipitated with the ARL6IP5, suggesting their interactions (Figure 5A). We also confirmed this interaction by detecting flag-ARL6IP5 after immunoprecipitating ATG12 (Figure 5B). All co-immunoprecipitation experiments were conducted in the ARL6IP5 overexpression condition, and hence we decided to conduct further study in the presence of basal ARL6IP5 expression level. We found that ATG12 co-immunoprecipitated with basal ARL6IP5 (Figure 5C), indicating that these two proteins interact under normal cellular physiological conditions. We also checked the interaction of Rab1 and ARL6IP5, as previously reported [35], using co-immunoprecipitation (Figure 5D). Interaction was also confirmed by the co-immunofluorescence (Figure 5E). We found co-localization of the red (ARL6IP5) and green (ATG5/ATG12 (80 ± 23%, *n* = 12, and 86 ± 18%, *n* = 12, respectively) in the cells (Figure 5F). It was previously reported that ATG12 is a highly unstable protein when it is not combined with ATG5 and is susceptible to ubiquitination and proteasomal degradation [36]. ATG12 is a limiting factor in the process of autophagy, stabilization of ATG12 may lead to increased autophagy. Since we know that elevating cellular levels of ATG12 through ARL6IP5 overexpression leads to increased autophagy, it is interesting to compare the levels of ATG12 ubiquitination under the overexpressing and downregulating conditions. We immunoprecipitated the ATG12 in the ARL6IP5 overexpressing condition and found a 56 ± 4%, *p* = 0.0032 reduction in ubiquitination and a 169 ± 19%, *p* = 0.0001 increase in ubiquitination after siRNA knockdown of ARL6IP5 as compared to control (Figure 5G–I). Taken together, these data suggest that ARL6IP5 interacts with ATG12 and prevents its ubiquitination, resulting in the upregulation of autophagy.

## 3. Discussion

The extended studies we conducted on understanding the role of ARL6IP5 in Parkinson’s disease and aging revealed it was downregulated in the brain as it aged, and this downregulation was even more pronounced in PD animal, human, and cellular models. These findings were sufficient to hypothesize the pivotal role of ARL6IP5 in PD. ARL6IP5 is a membrane resident protein of ER, mainly studied in the field of cancer [16,17,18]. No literature is available to predict its role in synucleopathy and neuronal autophagy. To decipher the role of ARL6IP5 in PD, we first studied the effect of overexpression and knockdown of ARL6IP5 in PD cellular model. The beneficial and pro-survival effects of ARL6IP5 overexpression in the cellular model of PD confirmed its pivotal role in neuronal physiology. This suggests that α-synuclein-mediated downregulation of ARL6IP5 might be an initial step involved in disease pathogenesis. The detailed mechanistic studies to understand its beneficial effect revealed that overexpression of ARL6IP5 significantly reduced toxic aggregates, reduced toxicity in LDH assay, and induced differentiation in the cellular model of PD (Figure 2A–D). The RTKs array was performed to understand their involvement in ARL6IP5-mediated neuronal survival. The analysis indicated activation of TAM pathway specifically upon ARL6IP5 overexpression (Figure 2F–H). TAM pathways play a key role in neuron survival and homeostasis [37], and its induction in ARL6IP5-mediated neuronal survival indicates its regulation by ARL6IP5. We further ventured to find out the intracellular pathway involved in removing toxic α-synuclein aggregates. The α-synuclein aggregates, being cytosolic, are removed by all cellular degradation pathways viz proteasomal, chaperon-mediated, and autophagy, depending on the size of the aggregates. We did not find significant changes in proteasomal- and chaperon-mediated degradation pathways (Appendix A). Interestingly, the process of autophagy, which was significantly reduced in A53T cellular model of PD, was augmented significantly upon ARL6IP5 overexpression, indicating its role in autophagy regulation. As we mentioned above, disruption/deregulation of the process of autophagy in neurons is linked to degeneration of dopaminergic neurons, leading to PD. Thus, ARL6IP5-mediated induction of autophagy and α-synuclein mediated downregulation of ARL6IP5 in PD constitute key findings of our study.

We extended our study to understand the ARL6IP5 mediated regulation of macro-autophagy. We were stunned to find that overexpressing ARL6IP5 in the SH-SY5Y cells induces autophagy, which is comparable to the autophagy induced by the standard chemical inducers of autophagy (Figure 3A–D). To confirm the phenomenon, we knockdown the ARL6IP5 and found a significant reduction in the basal level and chemically induced autophagy (Figure 3G–J). In ARL6IP5 overexpressing condition, we found that the level of free ATG12 is increased, which is highly susceptible to degradation [36]. ATG12 interacts with ATG5 and ATG16L and contributes to the elongation of autophagosome membrane [38]. In overexpressing and downregulating conditions of ARL6IP5, we found that free ATG12 increased (Figure 4E,F) and decreases (Figure 4G,H), respectively, and significantly, suggesting that ARL6IP5 stabilized ATG12. Post-translational modifications like ubiquitination plays an important role in the regulation of ATG12 cellular degradation. In the next experiment, we confirmed that ARL6IP5 inhibits the ubiquitination of ATG12 and hence its degradation (Figure 5G–I), which is the cause of elevated autophagy.

ARL6IP5 negatively regulates the Rab1 [35], function of transporting proteins from ER to Golgi. Rab1 plays an important role in the formation of the phagophore assembly site (PAS) [39], where the formation of autophagic membrane start. We have shown that overexpression of ARL6IP5 induces autophagy, suggesting that ARL6IP5 changes role of Rab1 from protein transportation to autophagy induction. Thus, ARL6IP5 is a downstream regulator of autophagy and might play a pivotal role in autophagosome membrane formation. This hypothesis also suggests that ARL6IP5 can induce autophagy in more than one way, which warrants independent investigation.

## 4. Material and Methods

### 4.1. Plasmids

Flag-ARL6IP5 plasmid was a kind gift from professor Jianwei Zhou, Department of Molecular Cell Biology and Toxicology, School of Public Health, Nanjing Medical University, 101 Lonamian Avenue, Jiangning District, Nanjing, 211166, China, EGFP-LC3BII plasmid was a kind gift from Dr. Dhiraj Kumar from ICGEB, New Delhi India and GFP-RFP-LC3BII (Addgene, Watertown, MA, USA # 84573), α-synuclein (Addgene #51437), GFP-A53T (Addgene #40823), pcDNA 3.1 vector (Invitrogen, Waltham, MA, USA V79020), siRNA targeting ARL6IP5(human) (Sigma, St. Louis, MO, USA #EHU041951)

### 4.2. Protein Extraction and Estimation

After the completion of the experiment, cells were lysed using 70 μL of lysis buffer (Promega, Madison, WI, USA, A1731) in each well. Protein concentration was measured after adding Bradford reagent (Thermo-Fisher Scientific, Waltham, MA, USA, 23236) and reading absorbance at 595 nm on a microplate reader (Infinite M200 ProMicroplate Reader TECAN, Seestrasse, Mannidorf, Switzerland).

### 4.3. Microscopy

Cells were fixed after the completion of the experiment using 4% PFA, after a wash, cells were covered by mounting media containing DAPI (Thermo-Fisher, P36961) and coverslip. Images were taken using a confocal microscope (ZEISS LSM 500, Oberkochen, Baden-Württemberg, Germany) at 63× magnification.

### 4.4. Reagents

Sodium bicarbonate (Sigma #S6014), L-glutamine (100× (Gibco #25030081), Trypsin EDTA (0.25%) (Gibco, Waltham, MA, USA #25200056), Opti-MEM (Gibco #31985062), 6-Hydroxydopamine hydrobromide (Sigma-Aldrich #H116), MG132 (Calbiochem, Burlington, MA, USA, #474790), Baf A1 (CST, Danver, MA, USA #54645), Chloroquine diphosphate salt (Sigma #C6628), Rapamycin (Calbiochem #553210), Methyl β-cyclodextrin (Sigma #C4555), Bovine Serum Albumin (HiMedia, Thane, Maharashtra, India #MB083), Lactate dehydrogenase activity assay kit (Sigma-Aldrich #MAK066), RTK kit (R&D Systems, Minneapolis, Min, USA, ARY001B), poly L-lysine (Sigma #P3513), paraformaldehyde (Sigma # 158127).

### 4.5. Antibodies

Antibodies against ARL6IP5 (Sigma#HPA015540, 1:1000 dilution), α-synuclein (CST #2628), GAPDH (sc-47724), Ubiquitin (CST #3933), HSC 70 (CST #8444), Beta-actin (Sigma #A5441), LC3A/B (MBL #M186-3, 1:1000 dilution), p62 (CST #8025), LC3IIB (CST #2775), Flag (Sigma #F7425), ATG 3(CST #3415), Rab1 (CST #12375), ATG5 (CST #8540), ATG12 (CST #2010), Rab8 (CST #6975)

### 4.6. In-Vivo Mouse Model Study

**A53T α-synuclein transgenic line** (Strain #: **004479**) and wild-type C57BL/6J (Strain #:**000664**) mice were procured from Jackson Laboratories and bred and maintained by the National Institute of Immunology small animal facility. Homozygous SNCA and wild-type littermates were used in this study. Animal handling and treatment strategies, including experiments, were carried out according to the standard guidelines of the Institutional Animal Ethics Committee (IAEC). Animal usage protocols were approved by the IAEC (Project # 345/14). Mice were anesthetized using xylazine (10mg/kg) plus ketamine(100mg/kg) and perfused transcardially using 0.1 M PBS, pH 7.4, containing 1% protease and phosphatase inhibitors (Sigma-Aldrich), following 4% (*w*/*v*) paraformaldehyde in 0.1 M PBS for fixation.

### 4.7. Cell Culture

Undifferentiated human neuroblastoma SH-SY5Y cells were maintained in high glucose DMEM medium supplemented with 10% fetal bovine serum (FBS) and 1% and 2% glutamine. Cells were grown at 37 °C in a humidified CO_2_ incubator with 5% CO_2_. Cells were tested for mycoplasma contamination using EZ-PCR Mycoplasma test kit (Biological Industries, Beit-Haemek, Israel). Cells were found to be negative for mycoplasma contamination.

### 4.8. Transfection

SH-SY5Y cells were seeded in 6-well plate and proceeded further for transfection and knockdown experiments after attending the desired confluency. Transient transfection was carried out with 2 μg of plasmid DNA using Lipofectamine with LTX plus (Thermo-Fisher Scientific, A12621) reagent by the manufacture’s protocol. The knockdown experiments were carried out with ARL6IP5 siRNA using Lipofectamine RNAi-MAX (Thermo-Fisher, 13778100) transfection reagent by the manufacture’s protocol.

For generation of stable cell lines, SH-SY5Y cells were transiently transfected, and single cells were selected for protein expression by drug-resistance (neomycin) or fluorescence-activated cell sorting. Stable cell lines generated were characterized by immunoblotting.

### 4.9. Protein Extraction and Estimation

Cells were lysed with cold RIPA lysis buffer (Sigma- Aldrich #R0278) supplemented with 1X protease inhibitor cocktail (Sigma, P8340) and phosphatase inhibitor cocktail (Sigma, P0044). Post-nuclear lysates were subjected to protein estimation via the Pierce^TM^ BCA protein assay kit (ThermoScientific #23225) using a TECAN Infinite M–200plate reader (Tecan Group Ltd., Mȧṅnedorf, Switzerland) containing MagellanTM data analysis software version 6.6.0.1; Tecan.

### 4.10. Immunoblotting

In Laemmli loading buffer [5% -mercaptoethanol, 0.05% bromophenol blue, 75 mM Tris HCl (pH 6.8), 2% SDS, and 10% glycerol], an equal amount of protein sample (50 g) was heat denatured at 37 °C for 30 min or 95 °C for 10 min. Protein samples were separated using 10%, 12%, and 14% SDS-PAGE (BioRad, Hercules, CA, USA). Resolved proteins were electroblotted on a 0.45 μm nitrocellulose membrane (MDI Membrane Technology, Ambala, India) in a glycine/methanol transfer buffer (20 mM Tris-base, 150 mM glycine, and 20% methanol) using the PROTEAN Mini Cell system (BioRad). The protein blots were blocked for 60 min and cut into 2–3 stripes (according to the molecular weight of the protein that needs to be probed) and again blocked for 30 min at room temperature with 5% BSA in 1× TBST. Blots were probed overnight at 4 °C with respective primary antibodies, followed by three 10-min washes with 1× TBST. Secondary antibody incubation was carried out with HRP-conjugated IgG goat anti-rabbit IgG and IgG goat anti-mouse for 90 min at room temperature. The membrane was washed three times with an interval of 10 min each, and then bands were detected with Clarity Western ECL substrate (Bio-Rad) in the LAS-4000 Fujifilm chemiluminescent gel documentation unit (GE Healthcare Life Sciences, Piscataway, NJ, USA). To normalize protein loading control, mouse anti-actin, and anti-GAPDH antibodies were used. Protein bands were analyzed by Multi Gauge Image Reader software version 2.0 (Fujifilm, Tokyo, Japan).

### 4.11. Immunohistochemistry

The paraffinized human midbrain sections of Parkinson’s disease and a control sample were obtained from the Brain bank of NIMHANS, Bangalore, India, with human ethics committee approval. The sections were deparaffinized, and optimized antigen retrieval (AR) was carried out with heat-mediated AR in citrate buffer (pH 6.0), followed by permeabilization with 0.2% triton X-100 in 1× PBS. The sections were washed three times with 1× PBS at 5-min intervals. Blocking was carried out with 10% goat serum for 90 min followed by primary antibody incubation at 4 °C in a humid chamber. Sections were again washed and incubated with Alexa fluor tagged secondary antibody for 90 min at room temperature. After three consecutive washes with 1× PBS, cells were treated with mounting media containing DAPI (ThermoFisher, P36961). Images were scanned using a confocal microscope (Zeiss LSM 980) at 63× magnification.

For mouse samples, SNCA transgenic and wild-type mice PFA fixed brains were cut into 10 µm thick sections using cryotome. Sections were heated in citrate buffer (pH = 6) for 15 min for antigen retrieval. Sections were blocked in 5% BSA at room temperature for 1 h. Sections were then washed thrice with PBS-T and incubated overnight in primary antibody (1:100) at 4 °C. Sections were again washed thrice with PBS-T and incubated in secondary antibody (1:100) for 1 h at room temperature. Sections were then washed thrice with PBS-T and covered with mounting media with DAPI (Thermo-Fisher, P36961) and coverslip. Slides were stored at 4 °C until microscopic analysis.

### 4.12. Immunofluorescence

Cells were seeded on poly L-lysine-coated coverslips. After reaching the required confluency, cells were transfected and fixed after the completion of the experiment using 4% (wt/vol) PFA for 15 min at room temperature, followed by three washes of 1× PBS for each of the next 5 min. Then, cells were covered by mounting media containing DAPI (ThermoFisher, P36961). Images were taken using a confocal microscope (ZEISS LSM 510) at 63× magnification.

### 4.13. Immunoprecipitation

After completion of the experiment, 200 µg of total protein from the cell lysate was incubated overnight on a slow rotation with 2 µg of primary antibody at 4 °C. Then, 40 µL of protein-A agarose beads were added to the lysate and incubated for another 2 h on slow rotation at room temperature. Lysate was centrifuged for 1 min at 1000 rpm to settle beads, and pelleted beads were washed three times using 1× PBS. Finally, SDS buffer was added to the pellet and heated at 95 °C for 10 min to remove all the protein attached to the antibody and further proceed for the immunoblotting. Total lysate was also utilized to run the western blot for checking the expression of the housekeeping gene GAPDH as a means of verifying equal protein loading.

### 4.14. Lactate Dehydrogenase Activity Assay

Cells were seeded in a 6-well plate, and co-transfection was carried out for 36 h. The supernatant fractions were collected and subjected to an assay for lactase dehydrogenase activity using the manufacturer’s protocol.

### 4.15. RTK Assay

The RTK assay was carried out using a human phospho-RTK array kit (Catalog no.# ARY001B). The assay was carried out according to manufacturer’s protocol.

### 4.16. Statistical Analysis

The data represented here are the means of three independent experiments, and the error bars represent the standard deviation (SD) between the experiments. Statistical analysis was performed using GraphPad Prism 8 (San Diego, CA, USA). Two tailed student *t*-tests and a one-way ANOVA using the Bonferroni post hoc method were applied to determine the significance among groups as indicated in the figures or figure legends. Statistical significances were represented by mentioning the *p-*values.

## 5. Conclusions

The significant decrease in the ARL6IP5 in PD and aging brain suggest its role in neurodegeneration, including PD pathogenesis. We have clearly shown that overexpression of ARL6IP5 ameliorates disease burden and improves neuronal survival. Our detailed analysis showed that ARL6IP5 is an ER resident protein playing a key role in autophagophore formation. Upregulation of ARL6IP5 can induce autophagy independently of the upstream initiation signal. Autophagy induction with standard methods also increased the ARL6IP5 level. Interestingly, the downregulation of ARL6IP5 downregulated autophagy even in the presence of autophagy inducers, establishing ARL6IP5 as a key protein involved in the process of pre-autophagophore formation. This is a key finding as the process of pre-autophagophore formation is not well defined. This shows the potential of ARL6IP5 as a target for diseases where autophagy is deregulated.

## Figures and Tables

**Figure 1 ijms-24-10499-f001:**
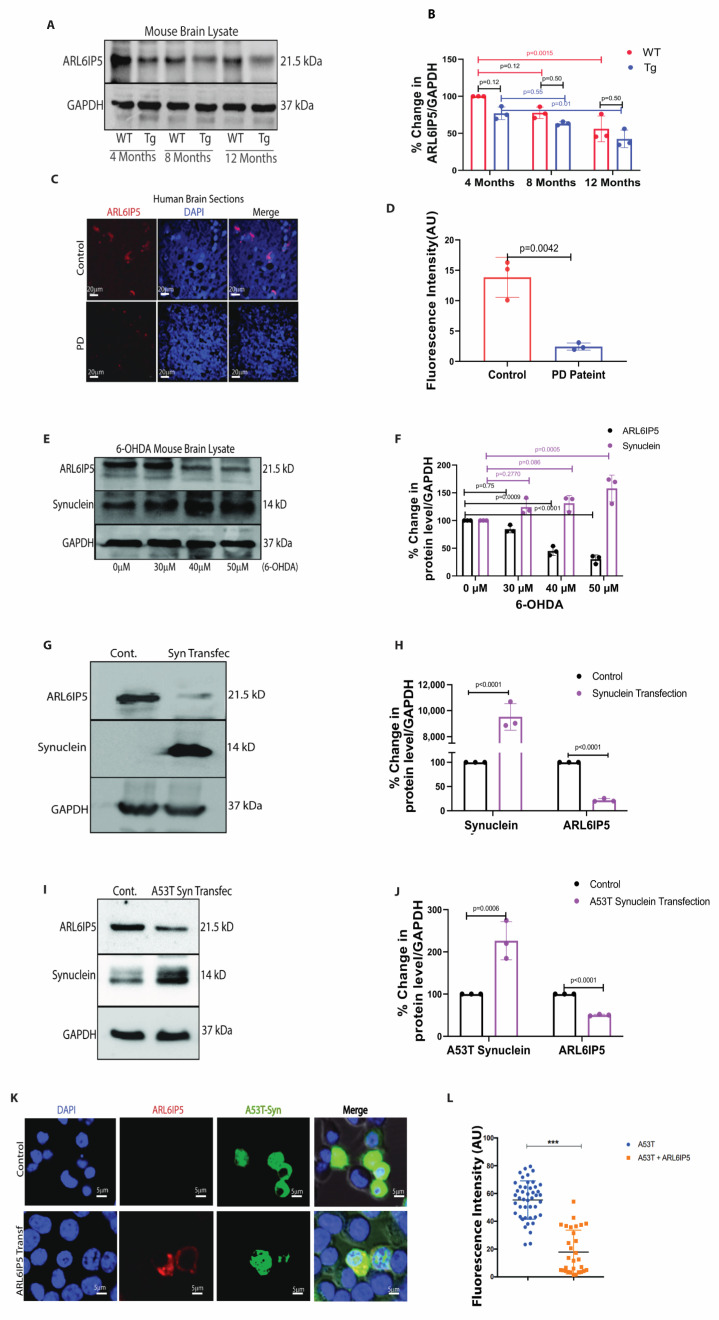
**ARL6IP5 level decreases with age and PD**. (**A**) Immunoblot showing age-dependent protein level of ARL6IP5 in WT and Tg mice brains. (**B**) Densitometric analysis of ARL6IP5 in the WT and Tg brains with respect to loading control GAPDH. *p-*value was calculated using two-way ANOVA with post hoc Tukey test. (**C**) ARL6IP5 (red) signals were detected in Parkinson’s patients and control sample’s mid-brain sections, as shown in representative IHC images. Scale bars, 20 μm. (**D**) Graph showing fluorescence intensity quantification. *p-*value was calculated using unpaired two-tailed Student *t*-test. (**E**) Immunoblots showing protein level of ARL6IP5 and α-synuclein upon treatment of 30, 40, 50 μM 6-OHDA for 30 days (**F**) Graph showing the densitometric analysis of ARL6IP5 level in 6-OHDA treated mice brains using GAPDH as a loading control. *p-*value was calculated using two-way ANOVA with Tukey post hoc method. (**G**) Immunoblots showing protein level of ARL6IP5 in α-synuclein overexpressed stable SH-SY5Y cells. (**H**) Densitometric analysis of ARL6IP5 in SH-SY5Y cells stably expressing α-synuclein as shown in graph with GAPDH as a loading control. *p-*value was calculated using one-way ANOVA with post hoc Tukey test. (**I**) Decreased ARL6IP5 protein level in A53T over expressed stable SH-SY5Y cells. (**J**) Graph showing densitometric analysis of ARL6IP5 in A53T overexpressed stable SHSY-5Y cells. *p-*value was calculated using one-way ANOVA with post hoc Tukey test. (**K**) Immunofluorescence representative image of SH-SY5Y cells expressing GFP-α-synuclein (green), stained for ARL6IP5 (red) in normal and ARL6IP5 overexpressing condition and nucleus (blue). Scale bars, 5 μm. (**L**) Graph showing the fluorescence intensity of A53T-α-synuclein in control and ARL6IP5 overexpressing condition, *n* = 40 cells were analyzed randomly. *p-*value was calculated using unpaired two-tailed Student *t*-test. *** *p* < 0.001. Results from three independent experiments are presented.

**Figure 2 ijms-24-10499-f002:**
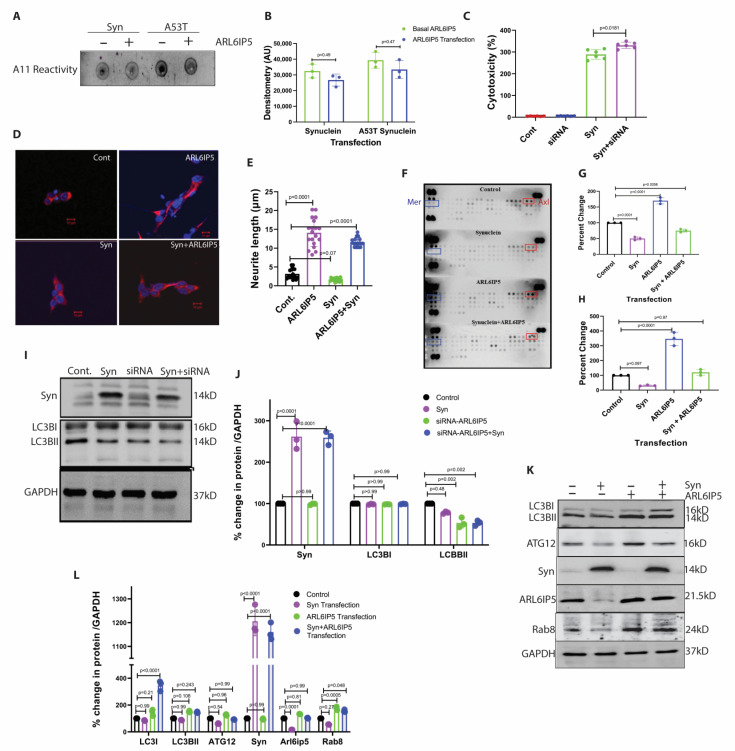
**ARL6IP5 overexpression reduces α-synuclein aggregates burden and induces differentiation.** (**A**) Representative dot-blot showing A11 reactivity for α-synuclein aggregates in the basal and overexpressed ARL6IP5 condition. (**B**) Densitometric analysis of A11 reactivity. *p*-value was calculated using one-way ANOVA with post hoc Tukey test. (**C**) Graph showing LDH activity in the supernatant samples from control, ARL6IP5 knockdown condition, in α-synuclein overexpressed stable cell line, and in knockdown ARL6IP5 and α-synuclein overexpressed condition. *p-*value was calculated using one-way ANOVA with Tukey test. (**D**) Representative confocal images of SH-SY5Y cells stained for MAP-2 (red). The nuclei are stained with DAPI (blue) for reference. n = 25. Scale bars, 10 μm. Neurite length was measured using the Image-J software version 1.53t. (**E**) Graph showing neurite lengths of SH-SY5Y cells in control, ARL6IP5 overexpressing, α-synuclein overexpressing, and ARL6IP5 and α-synuclein co-expressing conditions. *p-*value was calculated using one-way ANOVA with Tukey test. (**F**) Phospho-kinase array showing the Mer and Axl kinase levels in control, α-synuclein overexpressing, ARL6IP5 overexpressing, and α-synuclein and ARL6IP5 co-expressing conditions. (**G**,**H**) Graph showing the densitometric analysis of Mer and Axl levels, respectively. *p-*value was calculated using one-way ANOVA with post hoc Tukey test. (**I**) Representative blot showing the levels of autophagy proteins LC3BI, and LC3BII in control, α-synuclein overexpressing, ARL6IP5 knockdown, and ARL6IP5 knockdown condition with overexpressing α-synuclein, while GAPDH was used as the loading control. (**J**) Densitometric analysis graph of α-synuclein, LC3B1 and LC3BII. *p*-value was calculated using two-way ANOVA with post hoc Tukey test. (**K**) Representative blot showing the levels of LC3BI, LC3BII, ATG12, α-synuclein ARL6IP5, and Rab8 in control, α-synuclein overexpressing, ARL6IP5 overexpressing, and α-synuclein and ARL6IP5 co-expressing conditions. (**L**) Densitometric analysis of LC3BI, LC3BII, ATG12, α-synuclein, ARL6IP5, and Rab8. *p*-value was calculated using two-way ANOVA with post hoc Tukey test. Results from three independent experiments are presented.

**Figure 3 ijms-24-10499-f003:**
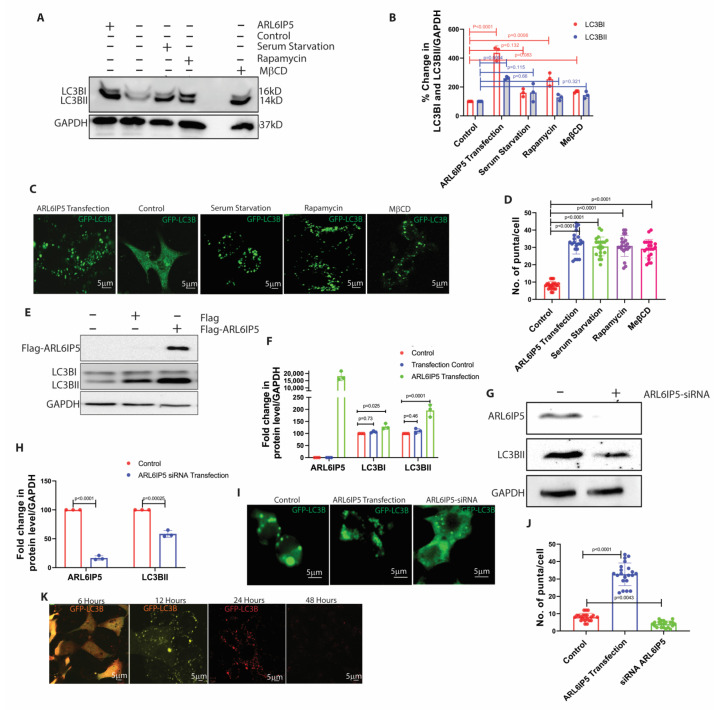
**ARL6IP5 regulates autophagy.** (**A**) Representative blot showing the levels of LC3BI, and LC3BII, in ARL6IP5 overexpressing, control, serum-starved, rapamycin-treated, and MβCD treated SH-SY5Y cells. GAPDH was used as the loading control. (**B**) Densitometric analysis of LC3BI and LC3BII is shown in image 3A. *p*-value was calculated using two-way ANOVA with post hoc Tukey test. (**C**) Representative confocal images showing the autophagy induction in the form of puncta formation in ARL6IP5 overexpressing, control, serum starved, rapamycin treated, and MβCD treated conditions. Scale bars, 5 μm. (**D**) Graph showing the number of puncta counted in ARL6IP5 overexpressing, control, serum-starved, rapamycin-treated, and MβCD treated conditions. *n* = 25 cells. *p*-value was calculated using one-way ANOVA with post hoc Tukey test. (**E**) Representative blot showing the levels of LC3BI and LC3BII in control and after Flag transfection, Flag-ARL6IP5 transfection conditions. (**F**) Densitometric analysis of the levels of LC3BI, and LC3BII in control, transfection control, and ARL6IP5 transfection conditions. *p*-value was calculated using one-way ANOVA with post hoc Tukey test. (**G**) Representative blot showing the level of LC3BII, in control and ARL6IP5 knockdown condition. (**H**) Densitometric analysis showing the level of ARL6IP5 and LC3BII in control and siRNA-ARL6IP5 knockdown condition. *p*-value was calculated using one-way ANOVA with post hoc Tukey test. (**I**) Representative confocal images showing the puncta formation levels in control, ARL6IP5 overexpressing, and ARL6IP5 knockdown conditions. *n* = 25. Scale bars, 5 μm. (**J**) Graph showing the number of puncta in individual cell in control, ARL6IP5 overexpressing, and knockdown conditions. *p*-value was calculated using one-way ANOVA with post hoc Tukey test. (**K**) Representative confocal images showing the various stages of the autophagy process using GFP-RFP-LC3B. First panel shows the protein in the cytoplasm giving yellow fluorescence (scale bar, 5 μm), second panel shows the formation of autophagosome (scale bar, 5 μm), third panel shows the fusion of autophagosome with lysosome (scale bar, 5 μm), and fourth panel shows the degradation of autophagosomes (scale bar, 5 μm). Results from three independent experiments are presented.

**Figure 4 ijms-24-10499-f004:**
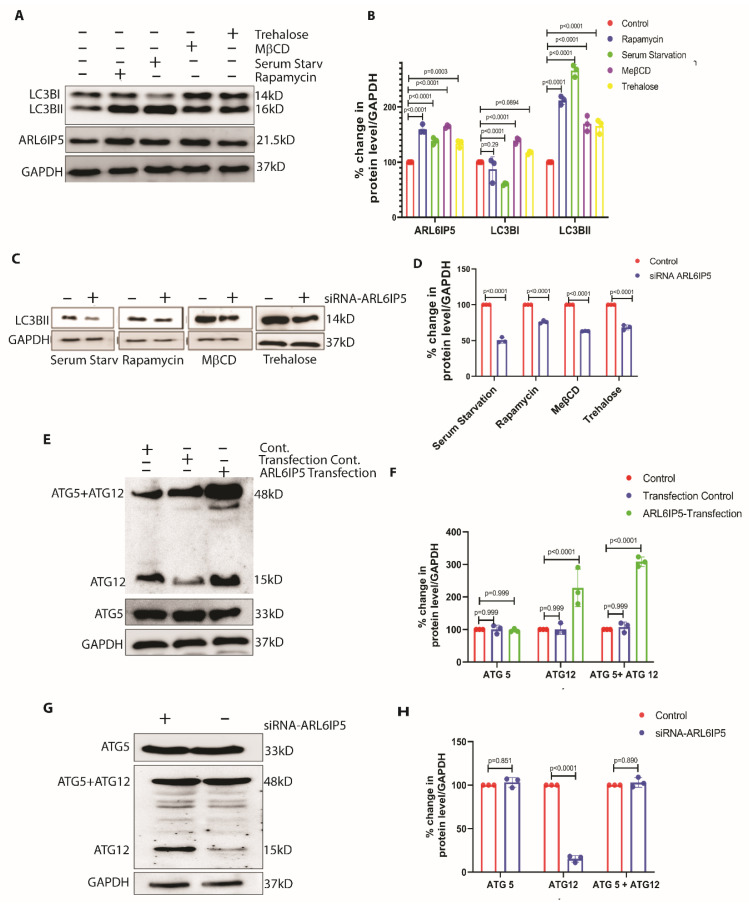
**ARL6IP5 regulates autophagy via ATG12.** (**A**) Representative blot showing the levels of LC3BI, LC3BII, and ARL6IP5 in control and after the induction of autophagy with Trehalose, MβCD, serum starvation, and rapamycin, respectively. (**B**) Graph showing the densitometric analysis of ARL6IP5, LC3B1 and LC3BII in control and with treatment with autophagy inducers shown in (**A**). *p*-value was calculated using two-way ANOVA with post hoc Tukey test. (**C**) Representative blot showing the levels of LC3BII, in control and ARL6IP5 knockdown conditions treated with autophagy inducers serum starvation, rapamycin, MβCD, and trehalose, respectively. (**D**) Densitometric analysis showing the levels of LC3BII, in control and ARL6IP5 knockdown conditions with autophagy inducers, as shown in (**C**). *p*-value was calculated using multiple *t*-test with two-stage linear step-up procedure. (**E**) Representative immunoblot showing the levels of ATG5 + ATG12, ATG12, and ATG5 in control, transfection control, and ARL6IP5 transfection conditions. (**F**) Densitometric analysis of the levels of ATG12, ATG5, and ATG12 + ATG5 from the blot (**F**). *p*-value was calculated using two-way ANOVA with post hoc Tukey test. (**G**) Representative blot showing the levels of ATG5 + ATG12, ATG5, and ATG12 in control and ARL6IP5 knockdown condition. (**H**) Densitometric analysis of the levels of ATG12, ATG5, and ATG5 + ATG12 in control and ARL6IP5 knockdown condition. Results from three independent experiments are presented. *p*-value was calculated using multiple *t*-test with two-stage linear step-up procedure.

**Figure 5 ijms-24-10499-f005:**
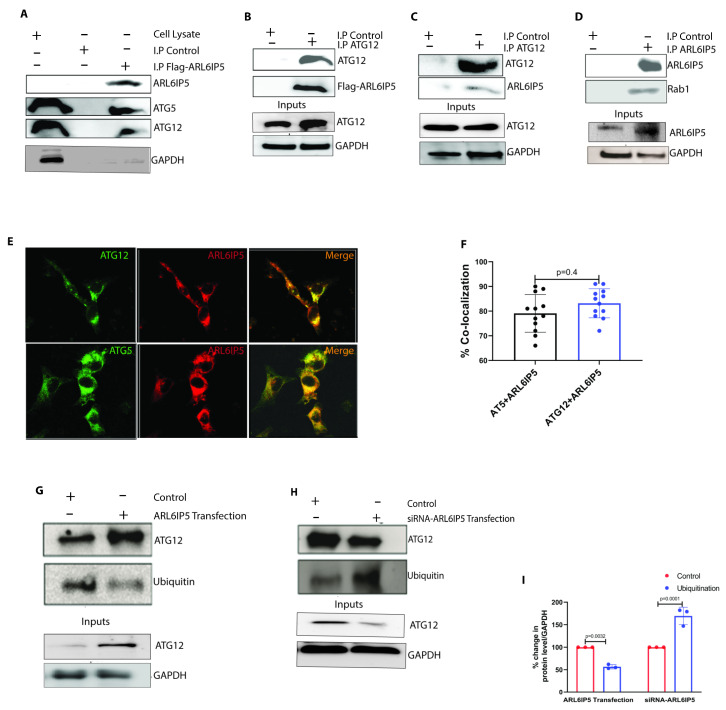
**ARL6IP5 interacts with ATG12 and prevents its ubiquitination**. (**A**) Representative blot showing the immunoprecipitation of ATG5, and ATG12 with ARL6IP5. (**B**) Representative blot showing the immunoprecipitation of overexpressed Flag-ARL6IP5 with ATG12 and GAPDH in the lysate. (**C**) Representative blot showing the immunoprecipitation of basal ARL6IP5 with ATG12 and GAPDH in the lysate. (**D**) Representative blot showing the immunoprecipitation of Rab1 with ARL6IP5 and GAPDH in the lysate. (**E**) Representative confocal images showing the co-localization (yellow) of ATG12 (green) and ARL6IP5 (red) (upper panel) and co-localization of ATG5 (green) and ARL6IP5 (red) (lower panel). Scale bars, 5 μm (**F**) Graph showing the percent of localization calculated by measuring the yellow area in the cell. *p-*value was calculated using unpaired two-tailed Student *t*-test. (**G**) Representative blot showing the levels of ATG12 ubiquitination in control and ARL6IP5 overexpressing condition. GAPDH was measured in the lysate. (**H**) Representative blot showing the levels of ATG12 ubiquitination in control and ARL6IP5 knockdown condition. GAPDH was measured in the lysate. (**I**) Densitometry analysis of the levels of ubiquitination in control and ARL6IP5 overexpressing and ARL6IP5 knockdown conditions. *p*-value was calculated using multiple *t*-test with two-stage linear step-up procedure. Results from three independent experiments are presented.

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
