# Peer review of "ARL6IP5 Ameliorates α-Synuclein Burden by Inducing Autophagy via Preventing Ubiquitination and Degradation of ATG12"

_ijms, 2023, doi:10.3390/ijms241310499_

Round 1

Reviewer 1 Report (Previous Reviewer 3)

Manuscript title: ARL6IP5 ameliorated synuclein burden by inducing autophagy via preventing ubiquitination and degradation of ATG12.

Thanks very much for the authors addressing most of my concerns. I still have some minor comments prior to publication.

1. Figure 1B, I meant the difference of ARL6IP5/GAPDH between WT and Tg in each group is not significant as the p=0.12, 0.55 and 0.50. However, the WB in Figure 1A showed that the difference is big there. It is hard to convince people your data is reliable.

2. Figure 1C, ‘the blue laser of our confocal system was not functional’ is not a good reason to explain your missing data. Maybe you move that one to the supplement.

3. Figure 2C, unpaired two-tailed Student t-test is not suitable here either.

4. Figure 5, could you please add the input, in other words, check the expression of ARL6IP5, ATG12, Rab1 and reference gene using the lysates before you run the IP?

Author Response

Reviewer 2 Report (Previous Reviewer 2)

All the comments included in the revised version of manuscript

Author Response

Thank you for your valuable suggestions.

This manuscript is a resubmission of an earlier submission. The following is a list of the peer review reports and author responses from that submission.

Round 1

Reviewer 1 Report

 The data presented by Siddique et al in the current manuscript is very interesting, results are well explained. However, I have certain queries that needs to be answered before the acceptance of this manuscript. Overall, I would recommend minor revision.

1.       In the abstract “To address 14 this in our present study we identified a novel regulator (ARL6IP5) of neuronal autophagy and 15 showed that the level of ARL6IP5 decreases in the brain with age and in Parkinson’s disease in 16 mouse and human” It is very confusing whether authors have reported any work from human brain at least it not supported by the data presented in the current manuscript. Simply mention in mouse models and human cell models.

2.       Improve the resolution of the figures, in some of the figures it is hard to follow.

3.       Line 382- 384 “The extended studies we conducted on understanding the role of ARL6IP5 in Parkinson's disease and aging revealed it was downregulated in the brain as it aged, and this 383 downregulation was even more pronounced in PD animal, human and cellular models” references are missing. References are also missing at the several places in the manuscript.

4.       Use consistently either ARL6IP5 or Arl6ip5 throughout the manuscript.

5.       Provide the siRNA sequence used for knockdown in the manuscript.

6.       It is not very clear from the figure legends whether data represented is from cell or animal models, amend the legends to make it easy for the reader.

7.       Extend the discussion part.

8.       Scale bar is missing in all confocal microscopy images.

9.       Include some more latest references in the manuscript.

10.   Check your manuscript thoroughly for kinds of typos and grammatical errors.

Author Response

I kindly request you to find the attachment.

Reviewer 2 Report

Attached as a file.

Author Response

(The authors gave the same response as above.)

Reviewer 3 Report

Manuscript title: ARL6IP5 ameliorated synuclein burden by inducing autophagy via preventing ubiquitination and degradation of ATG12.

In this manuscript, the authors identified a novel regulator (ARL6IP5) of neuronal autophagy and showed that the level of ARL6IP5 decreases in the brain with age and in Parkinson’s disease in mouse and human. They also confirmed that in cellular model of PD. The study also showed that ARL6IP5 induced autophagy via preventing ubiquitination and degradation of ATG12. It is very important for better understanding the mechanism of neurodegenerative diseases. However, I have a few comments that need further attention prior to resubmission.

1.     On line 97, please add some information about 6-OHDA.

2.     Figure 1B, we could see that the expression of ARL6IP5/GAPDH at 12 months has no change between WT and Tg. However, the Figure 1A the bands tell a different story. Please update accordingly.

3.     Figure 1C, please add the stain of DAPI.

4.     Figure 1E, the expression profile of ARL6IP5 in human is like some scatter particles. But it looks like ARL6IP5 expressed everywhere in the mouse tissues in Figure 1C. Could you please discuss the differences?

5.     Figure 1M, please check the labels on the left of the IF images (arl6ip5 transfection and control). I think it is opposite.

6.     Figure 2F, it seems that some other RTKs changed too. In your study, you mainly discussed TAM receptor family kinases Axl and Mer. Could you please discuss whether other RTKs involved in the synuclein aggregates?

7.     Figure 2K, from the western blot, the overexpression of ARL6IP5 did not work well. Please update.

In your method, please add the information of antibody RAB8.

And it is not suitable to do statistical analysis using an unpaired two-tailed t-test here.

8.     Figure 3A, please explain the reason why there is a blank between lane 4 and lane 6.

9.     Figure 3E and G, you missed the labels and bands sizes.

10.  Figure 3I and J, the quantification and the IF are not consistent.

11.  Figure 4B, C and Figure 3A, B, they have some overlaps. I highly recommend merge them and keep only one.

12.  Could you please merge Figure 4F and 4H together (like Figure 4I) if they are from the same lysate?

13.  Figure 4I, the bands for ATG5 are overexposed. Please update.

14.  Figure 5, please update all the WB including the expression of reference gene like GAPDH.

15.  Figure 5A, why is there no expression of ARL6IP5 in the cell lysate here? Figure 1I and 1K, Figure 2K, there are expression of ARL6IP5 in the control group. It means that there is endogenous expression of ARL6IP5.

16.  Figure 5D, the figure and the figure legend are not consistent. Please verify it is Rab1 or ATG12.

Author Response

(The authors gave the same response as above.)
